# Surgical Extent for Oral Cancer: Emphasis on a Cut-Off Value for the Resection Margin Status: A Narrative Literature Review

**DOI:** 10.3390/cancers14225702

**Published:** 2022-11-21

**Authors:** Jeon Yeob Jang, Nayeon Choi, Han-Sin Jeong

**Affiliations:** 1Department of Otolaryngology, Ajou University School of Medicine, Suwon 16499, Republic of Korea; 2Department of Otorhinolaryngology-Head and Neck Surgery, Samsung Medical Center, Sungkyunkwan University School of Medicine, Seoul 06351, Republic of Korea

**Keywords:** mouth neoplasm, surgery, margins of excision, treatment outcomes

## Abstract

**Simple Summary:**

Upfront surgical resection with safe margins is a mainstay of treatment in oral cancers. The postoperative risk stratification of the resection margin is currently determined through surgical pathology according to a cut-off width of 5 mm. However, evidence to support the validity of this cut-off point of 5 mm is not strong, and was largely obtained from retrospective clinical studies. In this review, we summarize surgical concepts for oral cancer, postoperative risk stratification based on current guidelines and propose a dynamic cut-off value for postoperative risk stratification in oral cancer.

**Abstract:**

The optimal cut-off point of the resection margin was recently debated in oral cancer. To evaluate the current evidence of the dynamic criteria of the resection margin, a review of the available literature was performed. Studies were sourced from PubMed and EMBASE by searching for the keywords “mouth neoplasm”, “oral cancer”, “oral cavity cancer”, “oral squamous cell carcinoma”, “tongue cancer”, “margins of excision”, “surgical margin” and “resection margin”. We found approximately 998 articles on PubMed and 2227 articles on EMBASE. A total of 3225 articles was identified, and 2763 of those were left after removing the duplicates. By applying advanced filters about the relevance of the subjects, these were narrowed down to 111 articles. After the final exclusion, 42 full-text articles were reviewed. The universal cut-off criteria of 5 mm used for determining the resection margin status has been debated due to recent studies evaluating the impact of different margin criteria on patient prognosis. Of note, the degree of the microscopic extension from the gross tumor border correlates with tumor dimensions. Therefore, a relatively narrow safety margin can be justified in early-stage oral cancer without the additional risk of recurrence, while a wide safety margin might be required for advanced-stage oral cancer. This review suggests a surgical strategy to adjust the criteria for risk grouping and adjuvant treatments, according to individual tumor dimensions or characteristics. In the future, it might be possible to establish individual tumor-specific surgical margins and risk stratification during or after surgery. However, the results should be interpreted with caution because there is no strong evidence (e.g., prospective randomized controlled studies) yet to support the conclusions. Our study is meaningful in suggesting future research directions and discussions.

## 1. Introduction

Oral cancer leads to major morbidity and mortality rates among head and neck cancers, and accounts for over 370,000 new cases and 170,000 deaths per year worldwide [1]. The five-year survival rate of oral cancer patients remains limited (50–65%) [2]. An improved understanding of the tumor biology and clinical features has led to recent changes in the American Joint Committee on Cancer (AJCC) tumor–node–metastasis (TNM) classification, to incorporate the depth of invasion (DOI) and the extranodal extension of lymph node metastasis as major determinants of the staging system [3].

Upfront surgery, including wide excision with adequate margins in all three dimensions, is a mainstay of treatment, and should be offered as the initial treatment when tumors are operable [4,5]. Surgery includes the neck lymphatic basins even in clinically node-negative patients, and it was proven in recent randomized controlled trials to impart the survival benefit of prophylactic neck dissection in early-stage oral cancer (T1-2N0) [6,7]. Sentinel lymph node biopsy has been reported to accurately predict the presence of pathological lymph node metastasis in early-stage oral cancer, and it is considered an alternative to elective neck dissection where expertise for this procedure is available [8,9,10,11,12].

According to the current treatment guidelines of the National Comprehensive Cancer Network Clinical Practice Guidelines in Oncology (NCCN guidelines) [4], the criteria of clear resection margin is considered to be 5 mm, regardless of the primary tumor’s size or characteristics. However, the supporting evidence for a cut-off point of 5 mm is not strong and has recently been debated, especially in early-stage oral cancer. In addition, several researchers have suggested a differential clinical significance to mucosal versus deep margins. Thus, the criteria of clear resection margin have been challenged based on the individual tumor characteristics, while the universal cut-off criterion of 5 mm is still adopted in the current treatment guidelines.

In this review, we summarized surgical concepts for oral cancer, postoperative risk stratification and current guidelines with a particular focus on the cut-off point for the resection margin from the relevant literature. In addition, we proposed a so-called dynamic cut-off value (rather than a fixed one) for postoperative risk stratification in oral cancer management based on recent research.

## 2. Method

A database search using the online databases PubMed (https://pubmed.ncbi.nih.gov/) and EMBASE (https://www.embase.com/) was conducted on 30 August 2022 (Figure 1). The following keywords were used in the search: “mouth neoplasm”, “oral cancer”, “oral cavity cancer”, “oral squamous cell carcinoma”, “tongue cancer”, “margins of excision”, “surgical margin”, “resection margin”. The inclusion criteria were articles regarding the impact of surgical margin on survival. Exclusion criteria were studies not related, nonrelevant articles, unavailable full-text articles and articles with insufficient data.

## 3. Results

We found approximately 998 articles on PubMed and 2227 articles on EMBASE. A total of 3225 articles was identified, and 2763 of those were left after the removal of the duplicates. After applying advanced filters through an abstract review, 2632 nonrelevant articles were excluded. The remaining 111 full-text articles were assessed for eligibility. After the final exclusion (studies not related, nonrelevant articles, unavailable full-text articles and those with insufficient data), the resulting 42 full-text articles were reviewed. Among them, regarding the subject of the prognostic significance according to the cut-off points of resection margin length, 12 studies were selected for the final analysis (Table 1). Regarding the subject of the dynamic cut-off points, four articles were selected for the final analysis (Table 2).

## 4. Discussion

### 4.1. Surgical Concept for Oral Cancer

The primary aim of curative surgery for oral cancer is the complete removal of local or locoregional disease to achieve a long-term disease-free status. Along with this fundamental basis of surgical resection, the functional preservation of normal tissues around tumors is essential to maintain patient quality of life within oncological safety [13].

In addition to the complete removal of clinically overt lesions, the identification and removal of potentially malignant (premalignant) lesions or clinically undetectable (microscopic) diseases are important factors for disease control. For example, high-grade dysplasia around cancerous lesions is a common target for resection, although the extent of high-grade dysplasia is not a criterion for the current tumor (T) classification in oral cancer [14]. Similarly, the primary surgery includes the surgical removal of occult lymph node metastasis, even in patients without a clinically detectable lesion (elective lymph node dissection or sentinel lymph node biopsy) [6].

This surgical principle is valid for the resection of the primary site of oral cancer, which includes normal or normal-appearing surrounding tissues at a certain margin (length or width) from clinical (gross) cancer lesions (Figure 2). This is the so-called surgical safety margin. While there is no high-level evidence for a specific cut-off value of the surgical safety margin [15], its clinical importance and indispensability have been accepted by most surgeons to accomplish the surgical aim safely.

More practically, a surgical safety margin in oral cancer surgery is necessary due to the following reasons:

#### 4.1.1. Irregular Three-Dimensional Growth of Primary Tumors

The current surgical approach to primary tumors largely depends on a visual inspection, palpation and preoperative imaging, all of which estimate the tumor to be a three-dimensional ellipsoid sphere. However, pathological evaluations of surgical specimens frequently reveal irregular three-dimensional growth patterns of primary tumors, such as a “finger-like” pushing pattern, satellite tumor nodules located away from the main tumor and tumor budding in oral cancers [16]. There is a clear limitation in precisely delineating the three-dimensional growth of tumors and irregular local extension in either surface or depth [17]. Therefore, surgery includes normal surrounding tissues at a certain margin from the gross tumor, so that the tumors can be removed completely with surgery, even in cases with an irregular local tumor infiltration.

#### 4.1.2. Difficulty in the Delineation of Normal to Tumor Margin during Surgery

The greatest weakness of the current surgery is its inability to clearly identify tumor boundaries on the mucosal surface or in deep tissues during surgery. It is often not possible to reflect accurately the boundaries of pathological tumors with the naked eye or even using magnified views with a microscope or endoscope. The limitations of deep resection frequently result in a suboptimal margin status. Some researchers have attempted real-time imaging, such as ultrasonography during surgery [18,19] or in vivo imaging [20,21,22,23]; however, a common and broad application of real-time imaging to a practice requires further development and clinical research. In short, a surgical strategy for complete tumor removal can be accomplished under the current surgical approach only with a tumor resection that includes safety margins from the boundaries of presumed gross tumors.

#### 4.1.3. Removal of Potential (Future) Malignant Sources

Even though all gross or microscopic tumor cells are removed during surgery (no actual tumor cells remain at the tumor resection margin), it is necessary to remove the adjacent tissue that has the potential to recur for long-term tumor control of the disease. These lesions include normal-looking tissues with premalignant characteristics and molecular alterations [24]. Even when the boundary of the tumor can be identified accurately and pathologically, it is safer to remove a certain amount of the surrounding normal-looking tissue at the boundary of the tumor to prevent future cancer recurrence.

### 4.2. Resection Margin Status and Risk Stratification

According to the current treatment guidelines (NCCN guidelines) [4], the resection margin status is divided into three categories. These categories are the criteria to predict the risk of tumor recurrence and to determine the best adjuvant treatment.

(1)Clear margin: defined as the distance from the invasive tumor front of 5 mm or more from the resected margin on final histopathology;(2)Close margin: defined as the distance from the invasive tumor front to the resected margin less than 5 mm on final histopathology;(3)Positive margin: defined as carcinoma in situ or invasive carcinoma at the margin of resection.

However, the supporting evidence for a cut-off point of 5 mm for the resection margin status is not strong, and was mostly obtained from retrospective clinical studies or surgeons’ experiences [15]. Moreover, the current cut-off point (5 mm) applies to all tumor stages (T1–4), regardless of individual tumor characteristics. According to the current treatment guidelines [4], the representative adverse features after surgery are positive resection margins and the extranodal extension of lymph node metastasis. In addition, pT3–4, N2–3, perineural invasion and lymphovascular invasion are risk factors for disease recurrence. Intriguingly, a close surgical margin is included in the postoperative adverse features in the latest version of the NCCN guidelines [4].

To achieve 5 mm postoperative pathological margins, the surgical method generally requires a safety margin of approximately 10–15 mm. The distance (10–15 mm) of this on-site surgical margin is based on both the tumor characteristics of irregular growth and expected postresection tissue shrinkage (30–70%) [25,26].

### 4.3. Criteria for a Close Resection Margin and Its Clinical Significance

Many clinical studies have investigated the optimal resection margin or margin length (width) in oral cancer surgery [14,25,26,27,28,29,30,31,32,33,34,35,36,37,38,39,40,41]. A recent meta-analysis of the resection margin size in oral squamous cell carcinoma (SCC) suggested that a 5 mm pathologic margin is the minimum acceptable value [42]. However, the five studies included in the meta-analysis had heterogeneities in their definitions of a close/positive resection margin and the status of adjuvant therapy.

The unanimous conclusions across studies are that the risk of local recurrence increases in cases with a positive surgical margin and is reduced with negative (clear) surgical margins after oral cancer surgery [14,24,31,33,40,41,43]. However, the impact of close surgical margins on local recurrences is still under debate (Table 1) [14,27,33,40,41,43,44,45,46,47]. It has been argued that the cut-off point of close margins could be 1 mm [44], 1.6 mm (in oral and oropharyngeal cancer) [35], 2 mm (in buccal mucosa cancer) [30], 3 mm [27], 5 mm [42] or 7 mm [40]. Particularly in early-stage oral cancer, the best margin size is more controversial. A previous study of 295 patients with pT1/T2 oral cancer reported that the size of the resection margin did not influence the local control rate [48]. Another study analyzed 382 patients with cT1/T2 oral cancer and found no significant increase in the local recurrence rate with a close resection margin, compared with a clear resection margin [49].

In addition, many researchers have imposed differential clinical significance to mucosal versus deep margins. Involved and close resection margins were more common in deep margins than in mucosal margins [50]. For example, 37.8% of oral cancer patients had a deep resection margin of less than 5 mm after en bloc resection, while 24.4% had a close mucosal resection margin [35]. Notably, a study of 187 patients with oral SCC reported that the local recurrence rate (64.0%) of a close (<4 mm) deep resection margin was higher than the 45.2% recurrence rate of the close mucosal resection margin [51]. Thus, the close deep resection margins occur more frequently and elicit a detrimental impact on prognosis compared with close mucosal resection margins, suggesting the requirement for the cautious consideration of the tumor-specific deep resection margin.

### 4.4. Dynamic Cut-Off Values for Postoperative Risk Stratification

With advances in our understanding of tumor biology, several interesting results have been published regarding different criteria of close surgical margins.

In human papilloma virus (HPV)-positive oropharyngeal cancer, the cut-off point for the resection margin for risk stratification decreased to 1–2 mm (when compared with the traditional margin criteria of 5 mm), based on the different (favorable) behaviors of HPV-related cancers [52,53]. This finding suggests that the criteria of risk grouping and adjuvant treatments can be adjusted based on individual tumor biology, even in oral cancers.

Currently, there are several supporting pieces of evidence suggesting the need for a dynamic margin cut-off system in oral cancer (Table 2). First, the surgical safety margin highly depends on the microscopic extension from the gross tumor border. In microscopically infiltrative cancers, a large amount of surrounding tissue must be included to avoid residual cancer cells. An interesting finding was that the degree of microscopic extension from the gross tumor border was found to correlate with tumor dimensions in oropharyngeal, oral and hypopharyngeal cancers [54,55,56]. That is, when the tumor is small or in the early growth stage, the degree of local invasion around the tumor is small; when the tumor is large, it increases (Figure 3). Therefore, a relatively narrow safety margin (or resection margin) can be applied in early-stage oral cancer, while a wide safety margin would likely be required in advanced-stage oral cancer.

**Table 1 cancers-14-05702-t001:** Prognostic significance and local recurrence rate according to the cut-off points of resection margin length in surgically treated oral squamous cell carcinomas.

Published Article First Author	Publication Year	Study Design and Number of Patients	Cut-Off Points of the Resection Margin	Local Recurrence Rate	Disease-Free Survival	Overall Survival (5Y)
Loree [41]	1990	Retrospective single center (N = 398)	<5 mm			52.0%
			≥5 mm			60.0%
Sutton [33]	2003	Retrospective single center (N = 200)	Involved	55%		11.0%
			<5 mm	33%		36.0%
			≥5 mm	12%		60.0%
Weijers [34]	2004	Retrospective single center (N = 68)	≤5 mm >5 mm	6.7% 7.9% (No difference)		
Garzino-Demo [37]	2006	Retrospective single center (N = 245)	<5 mm			48.0%
			≥5 mm			65.0%
Binahmed [43]	2007	Retrospective single center (N = 425)	Involved			38.7%
			<2 mm			58.3%
			≥2 mm			68.4%
Liao [40]	2008	Retrospective single center (N = 827)	3–11 mm in 1 mm intervals	On multivariate analysis, resection margin ≤ 7 mm was significantly associated with decreased local disease control		
Nason [27]	2009	Retrospective single center (N = 277)	Involved		48.3%	38.6%
			<2 mm		48.5%	62.6%
			3–4 mm		69.5%	69.6%
			≥5 mm		70.5%	72.9%
Kurita [14]	2010	Retrospective single center (N = 148)	1 mm	33.3%		
			2 mm	11.1%		
			3 mm	33.3%		
			Severe dysplasia	42.9%		
			Mild/mod dysplasia	0.0%		
Tasche [44]	2017	Retrospective single center (N = 432)	Involved	44.0%		
			<1 mm	28.0%		
			1 mm	17.0%		
			2 mm	13.0%		
			3 mm	13.0%		
			4 mm	14.0%		
			≥5 mm	11.0%		
Singh [45]	2020	Retrospective single center (N = 451)	≤2 mm	88.9%		
			3–7 mm	49.8%		
			≥8 mm	35.3%		
Jain [46]	2020	Retrospective single center (N = 612)	Involved		57.7%	38.5%
			<2 mm		60.0%	60.0%
			2–4 mm		76.8%	66.7%
			≥5mm		72.1%	76.3%
Lin [47]	2021	Taiwan Cancer Registry (N = 15,654)	Involved			46.7%
			<1 mm			69.5%
			1 mm			66.0%
			2 mm			71.8%
			3 mm			73.9%
			4 mm			74.8%
			≥5 mm			76.1%

**Table 2 cancers-14-05702-t002:** Supporting evidence for the dynamic criteria of cut-off margin lengths according to tumor dimension in oral cancer.

**Tumor Variables** **(Reference No.)**	**Number of Patients**	**Findings**	** *p* ** **-Value**		** *p* ** **-Value**
Tumor status [55]	90	Microscopic infiltration			
T1–2		0.96 ± 0.54 mm			
T3–4		1.76 ± 1.20 mm	<0.001		
Depth of invasion [57]	100	PNI		LVI	
<4 mm		2%		10%	
≥4 mm		38%	<0.01	28%	0.02
MTR in deep resection [58]	501	Two-year locoregional control rate			
>0.3		94%			
≤0.3		87%			
Log MTR [59]	302	Five-year disease-specific survival			
>33%		HR 1			
≤33%		HR 2.48	<0.001		

HR: hazard ratio; LVI: lymphovascular invasion; MTR: margin to thickness ratio; PNI: perineural invasion.

Besides a tumor’s dimensions, the depth of invasion is also known to be associated with adverse tumor characteristics. Larson et al. [57] recently reviewed one hundred pathologic specimens of oral cancers and found that the depth of invasion of more than 4 mm showed significantly increased rates of adverse features, including perineural invasion and lymphovascular invasion (Table 2).

There is other supporting evidence that has attempted to address the question of whether small oral cancers require the same margin clearance as large tumors [58]. The authors evaluated the association between the ratio of the closest margin to tumor thickness with local control and survival in oral cancers [58]. In this study, a margin-to-tumor thickness ratio (MTR) of 0.3 was set as a cut-off point, and an MTR of ≤0.3 was a predictor of local failure and disease-specific death (Table 2). These results suggest that a wider deep resection margin is required as tumor thickness increases. Another study further evaluated the value of this dynamic margin criteria and found that a log MTR of <33% was a predictor of less favorable outcome in the disease-specific survival of oral cancer [59]. Taken together, recent evidence suggests the need for dynamic resection margin criteria according to a tumor’s dimensions or the depth of invasion in oral cancer.

However, the level of evidence to support our conclusion is low. There is still no standard tool for applying individual tumor characterization to surgery (especially the surgical margin of the primary tumor). Therefore, additional clinical and basic studies are needed in the future.

## 5. Conclusions

In summary, the current treatment guidelines adopt a cut-off point of 5 mm as a single reference value for postoperative risk stratification, without considering tumor characteristics, dimensions or clinical tumor stage. However, as our understanding of tumor biology increases, it could be possible to establish tumor-specific surgical margins (risk stratification) either during or after surgery.

## Figures and Tables

**Figure 1 cancers-14-05702-f001:**
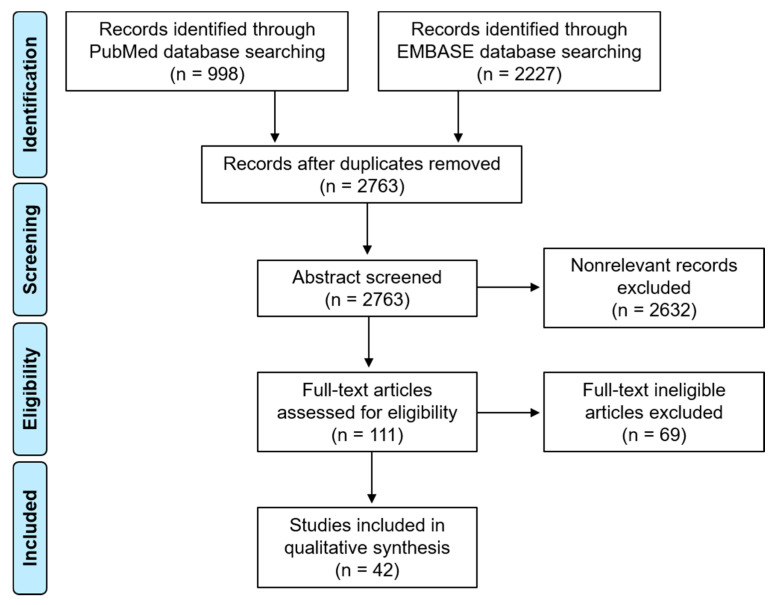
Flow chart for database search.

**Figure 2 cancers-14-05702-f002:**
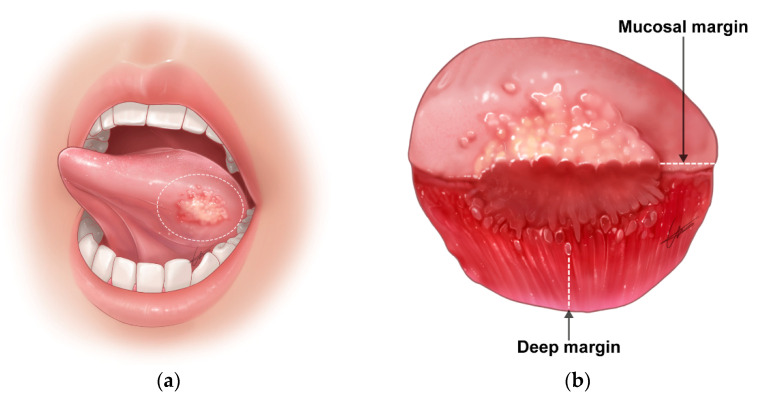
Resection of oral cancer with irregular three-dimensional growth. (**a**) Setting of appropriate safety margin in oral cancer surgery. Dotted line indicates mucosal safety margin; (**b**) surgical specimen that includes primary tumor and normal surrounding tissues. The resection margin includes mucosal and deep margins.

**Figure 3 cancers-14-05702-f003:**
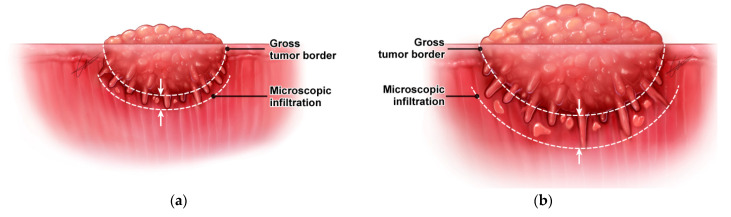
Microscopic infiltration depending on the gross tumor dimension. (**a**) A limited microscopic infiltration in a small-sized primary tumor; (**b**) an increased degree of microscopic infiltration in a large primary tumor.

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
