# Peer review of "Surgical Extent for Oral Cancer: Emphasis on a Cut-Off Value for the Resection Margin Status: A Narrative Literature Review"

_cancers, 2022, doi:10.3390/cancers14225702_

Round 1
Reviewer 1 Report (Previous Reviewer 3)
Dear editor,
The authors have incorporated all the sugestions and the text is much better now.
I recommend the manuscript for publication.
Sincerely yours,
Alena Medrado
Reviewer 2 Report (Previous Reviewer 2)
The previous comments have been addressed. Thank you
Reviewer 3 Report (Previous Reviewer 1)
Thanks for sincere response and correction to reviewer's comments.
This manuscript is a resubmission of an earlier submission. The following is a list of the peer review reports and author responses from that submission.
Round 1
Reviewer 1 Report
It is an interesting paper to thoroughly review the current evidences regarding the dynamic resection margin cut-off in oral cancer. Although the universal cut-off criteria of 5 mm have been adopted traditionally, recent evidences suggest the wide resection margin is unnecessary in early-stage diseases. The conflicting results may give a confusion to the clinicians in daily practices. Thus, it seems timely important aspects to thoroughly review the subjects.
Detailed comments
1. The word “oral cancer” or “oral cavity cancer” were mixed throughout the manuscript. It is recommended to select one representative word.
2. Although the manuscript was relatively clearly written, some grammatical errors should be re-evaluated (English proof reading).
3. Keywords might include Mesh terms as possible.
4. Overall quality of the manuscript is very good and offers an important progress in the field of head and oncology.
Author Response
AUTHOR RESPONSE
RE: Cancers-1978145
Surgical Extent for Oral Cancer: Emphasis on a Cut-Off Value for the Resection Margin Status: A Narrative Literature Review
Dear Editors and Reviewers,
Thank you for kind considerations of this article. We corrected it point-by-point and revised the manuscript following your recommendations. Corrected parts are marked with TRACK CHANGES.
We would be happy if the revised manuscript is more suitable to publication in the Cancers.
Best regards,
Han-Sin Jeong, the corresponding authors.
-------------------------------------------------------------------------------------------
Point-by-point response
Reviewer-1
It is an interesting paper to thoroughly review the current evidences regarding the dynamic resection margin cut-off in oral cancer. Although the universal cut-off criteria of 5 mm have been adopted traditionally, recent evidences suggest the wide resection margin is unnecessary in early-stage diseases. The conflicting results may give a confusion to the clinicians in daily practices. Thus, it seems timely important aspects to thoroughly review the subjects.
Answer: The authors appreciate for the reviewer’s valuable comments.
Detailed comments:
1. The word “oral cancer” or “oral cavity cancer” were mixed throughout the manuscript. It is recommended to select one representative word.
Answer: We have re-reviewed the manuscript and selected the representative word “oral cancer”.
2. Although the manuscript was relatively clearly written, some grammatical errors should be re-evaluated (English proof reading).
Answer: The authors agree with the suggestion. Additional English proof reading was performed by a professional editor.
3. Keywords might include Mesh terms as possible.
Answer: Keywords have been changed following the reviewer’s suggestion.
4. Overall quality of the manuscript is very good and offers an important progress in the field of head and oncology.
Answer: The authors thank for the positive comments.

Reviewer 2 Report
A very good idea but the presentation can be better.
1- Title: What is meant by systematic search? The term itself is not commonly used. Why did the authors prefer it rather than "systematic review"
2- The introduction is very broad and not specified. It should be re-written focusing on the issue of surgical margin in resection of oral cancer, the debates and the research gap the article tried to fill.
3- The methods section is very defective. What are the criteria for including & excluding an article? How was the conclusions built?
4- The results contain only the number of articles included. Very short. No statistical analysis performed. Nearly no "results"
5- Most of the discussion looks like a narrative review, not related to the results of the search. The limitations are not mentioned.
Overall, the article presents a mix of a narrative review and a systematic review, but failed to present any of them
Author Response
AUTHOR RESPONSE
RE: Cancers-1978145
Surgical Extent for Oral Cancer: Emphasis on a Cut-Off Value for the Resection Margin Status: A Narrative Literature Review
Dear Editors and Reviewers,
Thank you for kind considerations of this article. We corrected it point-by-point and revised the manuscript following your recommendations. Corrected parts are marked with TRACK CHANGES.
We would be happy if the revised manuscript is more suitable to publication in the Cancers.
Best regards,
Han-Sin Jeong, the corresponding authors.
-------------------------------------------------------------------------------------------
Point-by-point response
Reviewer-2
A very good idea but the presentation can be better.
Answer: The authors appreciate for the reviewer’s valuable comments.
1. Title: What is meant be systematic search? The term itself is not commonly used. Why did the authors prefer it rather than systematic review.
Answer: The authors agree with the reviewer’s comments. The term “systematic search” may be confused. We have removed the term and changed the title as “A Narrative Literature Review”.
2. The introduction is very broad and not specified. It should be re-written focusing on the issue of surgical margin in resection of oral cancer, the debates and the research gap the article tried to fill.
Answer: This is a valid point. We have added an additional paragraph in the introduction for focusing on the issue of surgical margin.
Correction: “According to the current treatment guidelines of the National Comprehensive Cancer Network Clinical Practice Guidelines in Oncology (NCCN guidelines) [4], the criteria of clear resection margin is considered as 5 mm regardless of primary tumor size or characteristics. However, the supporting evidences for a cut-off point of 5 mm are not strong and recently debated especially in early-stage oral cancer. In addition, several researchers have suggested a differential clinical significance to mucosal versus deep margins. Thus, the criteria of clear resection margin have been challenged based on the individual tumor characteristics while the universal cut-off criterion of 5 mm is still adopted in the current treatment guidelines.”
3. The methods section is very defective. What are the criteria for including and excluding an article? How was the conclusion built?
Answer: The authors appreciate for the reviewer’s critical comments and have added relevant contents in the methods section.
Correction:
Methods: “The inclusion criteria are articles regarding the impact of surgical margin on survival. Exclusion criteria are studies not related, non-relevant articles, unavailable full-text articles, and articles having insufficient data.”
Results: “Regarding the subject of the prognostic significance according to the cut-off points of resection margin length, 12 studies were selected for final analysis (Table 1). Regarding the subject of dynamic cut-off points, 4 articles were selected for final analysis (Table 2).”
4. The results contain only the number of articles included. Very short. No statistical analysis performed. Nearly no results;
Answer: The authors have changed the title as “A Narrative Literature Review”. In this study, no statistical analysis was performed, only a descriptive presentation was made by synthesizing existing research papers.
5. Most of the discussion looks like a narrative review, not related to the results of the search. The limitations are not mentioned.
Answer: This is a valid point. We have added relevant limitations in the discussion section.
Correction: “However, the evidence level to support our conclusion is low. there is still no standard tool for applying individual tumor characterization to surgery (especially surgical margin of the primary tumor). Therefore, additional clinical and basic studies are needed in the future.”
Overall, the article presents a mix of a narrative review and a systematic review, but failed to present any of them
Answer: The authors appreciate for the reviewer’s critical comments. We have selected the type of presentation as a narrative review and have tried to address the specific points.

Reviewer 3 Report
The manuscript intitled “Surgical Extent for Oral Cancer: Emphasis on a Cut-Off Value for the Resection Margin Status: A Systematic Search and Narrative Review" discusses a relevant topic. The text is clear and well-written. However, the authors should consider some important points to improve the manuscript.
1. Title: We suggest to use only “A Narrative Literature Review” at the end of the title;
2. Abstract: It is important to mention the limitations of the study at the end of the abstract;
3. Material and Methods: This section must be briefly described. The authors have to mention the criteria for selecting the manuscripts (relevance, date of publication). We do encourage the authors to add a table that summarize the main aspects of the included studies like sample, results and conclusions.
TThe manuscript discussed some important topics and its publication will surely contribute for the understanding of the resection margin status.
Author Response
AUTHOR RESPONSE
RE: Cancers-1978145
Surgical Extent for Oral Cancer: Emphasis on a Cut-Off Value for the Resection Margin Status: A Narrative Literature Review
Dear Editors and Reviewers,
Thank you for kind considerations of this article. We corrected it point-by-point and revised the manuscript following your recommendations. Corrected parts are marked with TRACK CHANGES.
We would be happy if the revised manuscript is more suitable to publication in the Cancers.
Best regards,
Han-Sin Jeong, the corresponding authors.
-------------------------------------------------------------------------------------------
Point-by-point response
Reviewer-3
The manuscript entitled “Surgical Extent for Oral Cancer: Emphasis on a Cut-Off Value for the Resection Margin Status: A Systematic Search and Narrative Review; discusses a relevant topic. The text is clear and well-written. However, the authors should consider some important points to improve the manuscript.
Answer: The authors appreciate for the reviewer’s valuable comments.
1. Title: We suggest to use only “A Narrative Literature Review” at the end of the title;
Answer: The authors agree with the reviewer’s comments and have changed the title following the reviewer’s suggestion.
2. Abstract: It is important to mention the limitations of the study at the end of the abstract;
Answer: This is a valid point. We have provided limitations of the study at the end of the abstract in the revised version of the manuscript.
Correction: “However, the results should be interpreted with caution because there is no strong evidence (e.g. prospective randomized controlled study) yet to support the conclusions. Our study is meaningful in suggesting future research directions and discussions.”
3. Material and Methods: This section must be briefly described. The authors have to mention the criteria for selecting the manuscripts (relevance, date of publication). We do encourage the authors to add a table that summarize the main aspects of the included studies like sample, results and conclusions.
Answer: The authors appreciate for the reviewer’s critical comments and have added relevant contents in the methods section.
Correction:
Methods: “The inclusion criteria are articles regarding the impact of surgical margin on survival. Exclusion criteria are studies not related, non-relevant articles, unavailable full-text articles, and articles having insufficient data.”
Results: “Regarding the subject of the prognostic significance according to the cut-off points of resection margin length, 12 studies were selected for final analysis (Table 1). Regarding the subject of dynamic cut-off points, 4 articles were selected for final analysis (Table 2).”
The manuscript discussed some important topics and its publication will surely contribute for the understanding of the resection margin status.
Answer: The authors appreciate for the reviewer’s encouragement.
